

# Deep learning-based electrocardiogram rhythm and beat features for heart abnormality classification

Annisa Darmawahyuni, Siti Nurmaini, Muhammad Naufal Rachmatullah, Bambang Tutuko, Ade Iriani Sapitri, Firdaus Firdaus, Ahmad Fansyuri and Aldi Predyansyah

Intelligent System Research Group, Faculty of Computer Science, Universitas Sriwijaya, Palembang, Indonesia

## ABSTRACT

**Background**. Electrocardiogram (ECG) signal classification plays a critical role in the automatic diagnosis of heart abnormalities. While most ECG signal patterns cannot be recognized by a human interpreter, they can be detected with precision using artificial intelligence approaches, making the ECG a powerful non-invasive biomarker. However, performing rapid and accurate ECG signal classification is difficult due to the low amplitude, complexity, and non-linearity. The widely-available deep learning (DL) method we propose has presented an opportunity to substantially improve the accuracy of automated ECG classification analysis using rhythm or beat features. Unfortunately, a comprehensive and general evaluation of the specific DL architecture for ECG analysis across a wide variety of rhythm and beat features has not been previously reported. Some previous studies have been concerned with detecting ECG class abnormalities only through rhythm or beat features separately.

**Methods**. This study proposes a single architecture based on the DL method with one-dimensional convolutional neural network (1D-CNN) architecture, to automatically classify 24 patterns of ECG signals through both rhythm and beat. To validate the proposed model, five databases which consisted of nine-class of ECG-base rhythm and 15-class of ECG-based beat were used in this study. The proposed DL network was applied and studied with varying datasets with different frequency samplings in intra and inter-patient scheme.

**Results**. Using a 10-fold cross-validation scheme, the performance results had an accuracy of 99.98%, a sensitivity of 99.90%, a specificity of 99.89%, a precision of 99.90%, and an F1-score of 99.99% for ECG rhythm classification. Additionally, for ECG beat classification, the model obtained an accuracy of 99.87%, a sensitivity of 96.97%, a specificity of 99.89%, a precision of 92.23%, and an F1-score of 94.39%. In conclusion, this study provides clinicians with an advanced methodology for detecting and discriminating heart abnormalities between different ECG rhythm and beat assessments by using one outstanding proposed DL architecture.

Corresponding author
Siti Nurmaini,
siti_nurmaini@unsri.ac.id,
sitinurmaini@gmail.com

# INTRODUCTION

Globally, heart abnormality deaths are projected to increase to 23.4 million, comprising 35% of all deaths in 2030 (*World Health Organization, 2018*). The clinical symptoms, electrocardiogram (ECG) pattern analysis, and measurement of important cardiac biomarkers are the current heart diagnostic cornerstone (*O'Gara et al., 2013*). However, such diagnosis is based on the invasive laboratory test and requires specific tools, cost, and infrastructures, such as trained clinical staff for inspecting blood and performing assays and a hematology analyser with biochemical reagents (*Cho et al., 2020*). For this reason, such assessments are difficult to use in remote healthcare monitoring or developing countries (*Makimoto et al., 2020*). Analysis of ECG patterns could help with early detection of life-threatening heart abnormalities and is considered for diagnosing patients' health conditions into specific grades, which can assist clinicians with proper treatment (*Siontis et al., 2021*). ECG measures the electricity of the heart, by analyzing each of electrical signal, it is possible to detect some abnormalities. In such conditions, ECG should allow continuous and remote monitoring.

Although the acquisition of ECG recordings is well standardized, human interpretations of ECG recordings vary widely. This is due to differences in the level of experience and expertise. To minimize these constraints, computer-generated interpretations have been used for various years. However, because this interpretation is based on predetermined rules and the limitations of the feature recognition algorithm, the interpretation results do not always capture the complexities and nuances contained in the ECG (*Siontis et al., 2021*). Based on this, the ECG by itself is often insufficient to diagnose several heart abnormalities. In myocardial infarction (MI), for example, is due to ST-segment deviation and may be occurred in other conditions such as acute pericarditis, left ventricular hypertrophy, left bundle-branch block, Brugada syndrome, and early repolarizations (*Wang, Asinger & Marriott, 2003*). Due of this, automatically diagnosing MI using a ruled based inference system from conventional ECG machine has a low reliability and by practice cardiologists are unable to diagnose it only from ECG record (*Daly et al., 2012*; *Cho et al., 2020*). Furthermore, the traditional methods for diagnosing heart abnormalities specifically from a 12-lead ECG are difficult to apply in wearable devices (*Walsh, Topol & Steinhubl, 2014*; *Cho et al., 2020*) and a wide variability in ECG morphology between patients causes major challenges.

The heart abnormalities analysis by using ECG signal processing can be conducted by using rhythm and beat feature. In the previous studies, such research has been proposed with several method. Deep learning (DL) is one type of artificial intelligence approach that can learn and extract meaningful patterns from complex raw data and recently has begun to widely used to analyze ECG signals for diagnosing an arrhythmia, heart failure, myocardial infarction, left ventricular hypertrophy, valvular heart disease, age, and sex with ECG alone and produce good result (*Darmawahyuni et al., 2021*; *Makimoto et al., 2020*; *Attia et al., 2019*; *Hannun et al., 2019*; *Kwon et al., 2020a*; *Kwon et al., 2020b*; *Yildirim, 2018*; *LeCun, Bengio & Hinton, 2015*). DL performs excellently over a relatively short period of time. The sophistication of DL has a much better ability to feature representation at an abstract

level compared to general machine learning. The DL model can extract a hierarchical representation of the raw data automatically and then utilize the last stacking layers to gain knowledge from complex features to the simpler ones (*Khan & Yairi, 2018*).

In the previous study, the ECG signal classification based on heart rhythm can be conducted with several features morphology of ECG signal like presenting ST-elevation and depression, T-wave abnormalities, and pathological Q-waves (*Ansari et al., 2017*). Moreover, a variety of ECG rhythm features, such as the R-R interval, S-T interval, P-R interval, and Q-T interval have been implemented to automatically detect heart abnormalities over the past decade (*Gopika et al., 2020*). Unlike an ECG rhythm, the efficiency classification of the irregular heartbeat , either faster or slower than normal, or even waveform malformation can be improve by using beat feature (*Khalaf, Owis & Yassine, 2015*). For heartbeat classification, ECG pattern may be similar for different patients who have different heartbeats and may be different for the same patient at different times. ECG-based heartbeat classification is virtually a problem of temporal pattern recognition and classification (*Zubair, Kim & Yoon, 2016*; *Dong, Wang & Si, 2017*). Based on the aforementioned instances, the variety of ECG signals with abnormalities must be handled specifically, either as an ECG rhythm or beat features.

Unfortunately, the challenge in analyzing the pattern of ECG signal is not limited to this. ECG signals have small amplitudes and short durations, measured in millivolts and milliseconds, respectively, and large inter- and intra-observer variability that influences the perceptibility of these signals (*Lih et al., 2020*). The analysis of thousands of ECG signals is time-consuming, and the possibility of misreading vital information is high. Automated diagnostic systems can utilize computerized recognition of heart abnormalities based on rhythm or beat to overcome such limitations. This could become the standard procedure by clinicians classifying ECG recordings.

Hence, the present study proposes a single DL architecture for classifying ECG patterns by using both rhythm and heartbeat features. Rather than treating ECG heartbeat and rhythm separately, we process both of them in the same framework. Hence, we only need a single DL architecture to classify the ECG signal with high accuracy. DL-based frameworks mainly include a stacked autoencoder (SAE), long short-term memory (LSTM), a deep belief network (DBN), convolutional neural networks (CNN), and so on. Among DL algorithms, we have generated a one-dimensional CNN (1D-CNN) model and showed promising results in our previous works (*Nurmaini et al., 2020*; *Tutuko et al., 2021*). In other works, 1D-CNN has also performed well for ECG classification, with overall performances ranging from 93.53% to 97.4% accuracy using rhythm (*Acharya et al., 2017*; *Wang, 2020*) and with overall 92.7% to 96.4% accuracy using beat (*Zubair, Kim & Yoon, 2016*; *Kiranyaz, Ince & Gabbouj, 2015*). For the pattern recognition technique, 1D-CNN is well known, as it integrates feature extraction, dimensionality reduction, and classification techniques utilizing several convolution layers, pooling layers, and a fully connected layer. Convoluted optimum features are derived and classified using feed-forward artificial neural networks using a fully connected layer with a learning framework for back propagation (*Li et al., 2019*). This study and the proposed approach make the following novel contributions:

- Proposes the generalization framework of deep learning for ECG signal classification with high accuracy in intra- and inter-patients' scenario;
- Develops a single DL-architecture for classifying ECG signal pattern based on both of rhythm and beat feature with simple segmentation;
- Validates the proposed framework with five public ECG datasets that have different frequency sampling with massive data; and
- Experiment with 24-class abnormalities found in the ECG signal, consisting of nine-class of ECG pattern based on rhythms feature and 15-class of ECG pattern based on beats feature.

The rest of this paper is organized as follows. Section 2 presents the materials and methods, which comprise ECG raw data and the proposed methodology for ECG rhythm and beat classification using 1D-CNN architecture. Section 3 presents the results and discussion. Finally, the conclusions are presented in Section 4.

## MATERIAL AND METHODS

### Data preparation

In this study, we use the public data set from PhysioNet (*Goldberger et al., 2000*). The ECGs data in the Physionet were collected from healthy volunteers and patients with different heart diseases. This database has already been published online by a third party and unrelated to the study. Consequently, there should be no concerns regarding the ethical disclosure of the information. To process the ECG signal pattern recognition, we utilize two segmentation processes, rhythm and beat. Therefore the experimental databases is divided into two cases, (i) for ECG rhythm classification utilize the PTB Diagnostic ECG (PTB DB) (*Bousseljot, Kreiseler & Schnabel, 1995*), the BIDMC Congestive Heart Failure (CHF) (*Baim et al., 1986*), the China Physiological Signal Challenge 2018 (*Liu et al., 2018*), the MIT-BIH Normal Sinus Rhythm (*Goldberger et al., 2000*); and (ii) for ECG beat classification utilize the MIT-BIH Arrhythmia Database (*Moody & Mark, 2001*).

A summary of each database is provided as follows:

- The PTB DB contains 549 records from 290 patients (209 men and 81 women). ECG signals were sampled at 1000 Hz. Each ECG record includes 15 signals measured simultaneously: 12 conventional leads (I, II, III, aVR, aVL, aVF, V1, V2, V3, V4, V5, V6) along with three ECG Frank leads (vx, vy, vz) in the .xyz file. For this study, only a single lead (lead II) was used. The database provides ECG normal and nine heart abnormalities, such as myocardial infarction, cardiomyopathy, bundle branch block, dysrhythmia, hypertrophy, myocarditis, and valvular heart disease.
- The BIDMC CHF database includes ECG recordings of about 20 h in duration from 15 patients with severe congestive heart failure conditions, that is, New York Heart Association (NYHA) Class 3 and 4. The database contains two ECG signals that are sampled at 250 samples per second.
- The China Physiological Signal Challenge 2018 database was collected from 11 hospitals and contains 12-lead ECG recordings lasting 6–60 s. All recordings were sampled at

500 Hz. The database provides three training sets, and all training sets were used in this study. However, only lead II was used for this study.

- The MIT-BIH Normal Sinus Rhythm includes 18 long-term ECG recordings that were found to have no significant arrhythmias. The database contains five male and 13 female patients. ECG recordings were sampled at 128 Hz.
- The MIT-BIH Arrhythmia Database use for beat classification. This study experimented with which was obtained from 47 different patients. It contained 48 half-hour excerpts from two-channel ambulatory ECG recordings. All ECG recordings were digitized at 360 samples per second. For the present study, we utilized ECG beat types of entire records. Most of the beats recorded in these databases have annotations associated with the type of beat or the events.

The total records, labels, and plots of the nine-class pattern for ECG rhythm classification are presented in Table 1 and Fig. 1, and the 15-class pattern for ECG beat classification were analyzed for this study are presented in Table 2 and Fig. 2.

## Proposed methodology of 1D-CNN

In this study, we present a methodology using single DL architecture to classify 25 classes of ECG signal pattern of based on rhythm and beat feature. Unlike others methodologies that treat beat and rhythm separately, our approach enables both forms to proceed on a single DL architecture. The methodology includes ECG signal denoising, beat and rhythm segmentation and classification. Using this approach, pattern abnormalities that occurred in the ECG, both in beats and rhythms, can be detected only using single architecture. We generalized the 1D-CNN architecture that was published in previous work (*Nurmaini et al., 2020*; *Tutuko et al., 2021*). The proposed methodology of the 1D-CNN generalized architecture is presented in Fig. 3. The general process in the methodology with standardize the evaluation process considering a clinical point of view. This standardization and defined the workflow to perform the evaluation to make sure the experiments are reproducible and comparable. Aiming to standardize, the general methodology including five main stages are, (i) data base selection from the ECG public database; (i) the pre-processing stage of ECG signals by eliminating various kinds of noise and artifacts using discrete wavelet transforms; (iii) the segmentation stage for ECG signal based on rhythm and beat. The ECG signal segmented to 2,700 and 252 nodes, respectively; (iv) 1D-CNN, as the feature extraction and classifier, learn the characteristics of each rhythm and beat episodes for ECG signal classification. and (v) the evaluation stage of the proposed model based on validation and testing data with accuracy, sensitivity, specificity, precision and F1-score.

1. **Database Selection** We have total of 168,472 rhythm episodes and 110,082 beat episodes as ECG features were used for training, validation, and an testing (as unseen data). The 1D-CNN architecture was used to classify the nine-class by using rhythms feature segmentation and 15-class of beats feature segmentation of the ECG signal. The information available from the single-lead ECG standard recordings included different signal lengths and frequency samplings (128, 250, 500, and 1000 Hz).

2. **Pre-processing** The ECG signal can become corrupted during acquisition due to different types of artifacts and interference, such as muscle contraction, baseline

**Table 1   ECG rhythm data description.**

| Dataset | Class | Label/ Abbreviation | Records |
|---|---|---|---|
| PTB diagnostic ECG | Bundle branch block | BBB | 17 |
| | Cardiomyopathy | C | 17 |
| | Dysrhythmia | D | 16 |
| | Health control | HC | 80 |
| | Myocardial hypertrophy | H | 7 |
| | Myocardial infarction | NU | 368 |
| | Myocarditis | M | 4 |
| | Valvural | VHD | 6 |
| BIDMC congestive heart failure | Congestive heart failure | HF | 10 |
| China physiological signal challenge 2018 | Left bundle branch block | BBB | 207 |
| | Right bundle branch block | | 1,695 |
| MIT-BIH normal sinus rhythm | Normal sinus (healthy control) | HC | 18 |

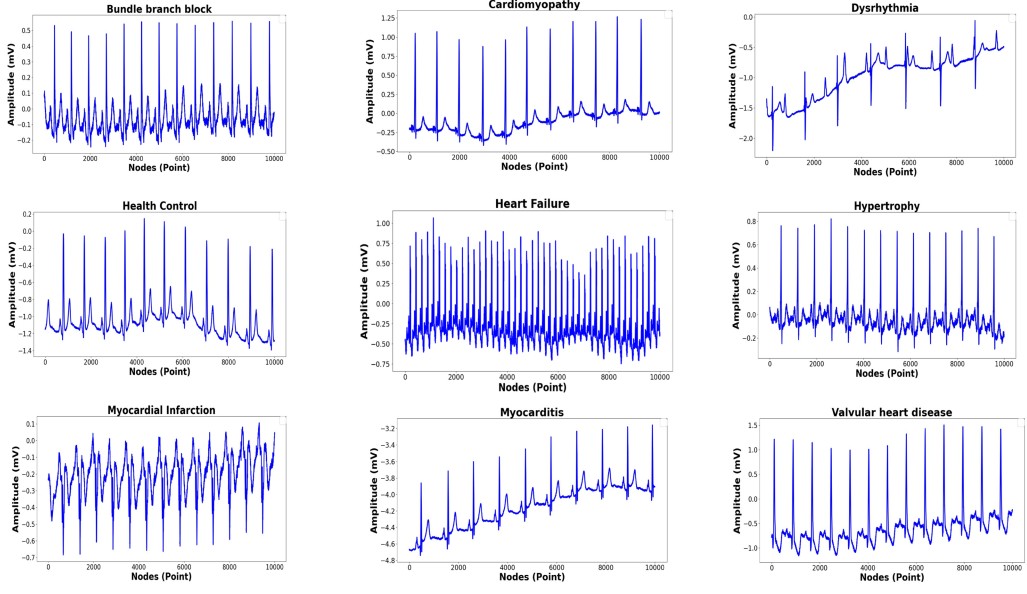

**Figure 1   Nine-classes of ECG-based rhythm classification.**

drift, electrode contact noise, and power line interference (*Sameni et al., 2007*; *Tracey & Miller, 2012*; *Wang et al., 2015*). To achieve an accurate analysis and diagnosis, undesirable noise and signals should be removed or deleted from the ECG by eliminating various kinds of noise and artifacts. This study implemented DWT as a frequently used denoising technique that offers a useful option for denoising ECG signals. This study also implemented some wavelet families for ECG signals, such as *symlets, daubechies, haar, bior*, and *coiflet*, to analyze which type of wavelet would obtain the best signal denoising result. Among them, based on the highest the signal

**Table 2  ECG beat data description.**

| Dataset | Class | Total beats |
|---|---|---|
| MIT-BIH Arrhythmia | Normal Beat (N) | 75,022 |
| | Atrial Premature Beat (A) | 2,546 |
| | Premature Ventricular Contraction (V) | 7,129 |
| | Right Bundle Branch Block Beat (R) | 7,255 |
| | Left bundle branch block beat (L) | 8,072 |
| | Aberrated atrial premature beat (a) | 150 |
| | Ventricular flutter wave (!) | 472 |
| | Fusion of ventricular and normal beat (F) | 802 |
| | Fusion of paced and normal beat (f) | 982 |
| | Nodal (junctional) escape beat (j) | 229 |
| | Nodal (junctional) premature beat (J) | 83 |
| | Paced beat (/) | 7,025 |
| | Ventricular escape beat (E) | 106 |
| | Non-conducted P-wave (x) | 193 |
| | Atrial escape beat (e) | 16 |

noise to ratio (SNR) results, the *symlet* wavelet was the best DWT parameter and was chosen for ECG signal denoising.

3. **ECG Signal Segmentation** The aim of ECG segmentation is to divide a signal into many parts with similar statistical properties, such as amplitude, nodes, and frequency. The presence, time, and length of each segment of an ECG signal have diagnostic and biophysical significance, and the various sections of an ECG signal have distinctive physiological meaning (*Yadav & Ray, 2016*). ECG signal segmentation may also be accurately analyzed. The process of ECG feature segmentation for rhythm and beat classification can be described as follows:

- ECG rhythm segmentation is the process to produce the features for the entire ECG signal recordings at 2,700 nodes without considering the different frequency sampling (128, 250, 500, and 1,000 Hz) for ECG rhythm classification. In our previous work (*Nurmaini et al., 2020*), we successfully segmented the length of AF episodes to 2,700 nodes. Therefore, for this study, we generated the features for nine-class of normal-abnormal ECG rhythm. The length of 2,700 nodes contained at least two R-R intervals between one and the next beat with different frequency samplings in all records. Furthermore, the 2,700-node segmentations might show more than two R-R intervals with a minimum frequency sampling of 128 Hz for the training, validation, and unseen set. As a result, the best ECG episodes were chosen from 2,700 nodes for segmentation. The process of ECG rhythm classification is illustrated in Fig. 4A. Figure 4A shows that all lengths of the ECG recordings have been segmented to each episode of 2,700 nodes. If the total nodes were less than 2,700 nodes, we added the zero-padding technique, which involved extending a signal with zeros.

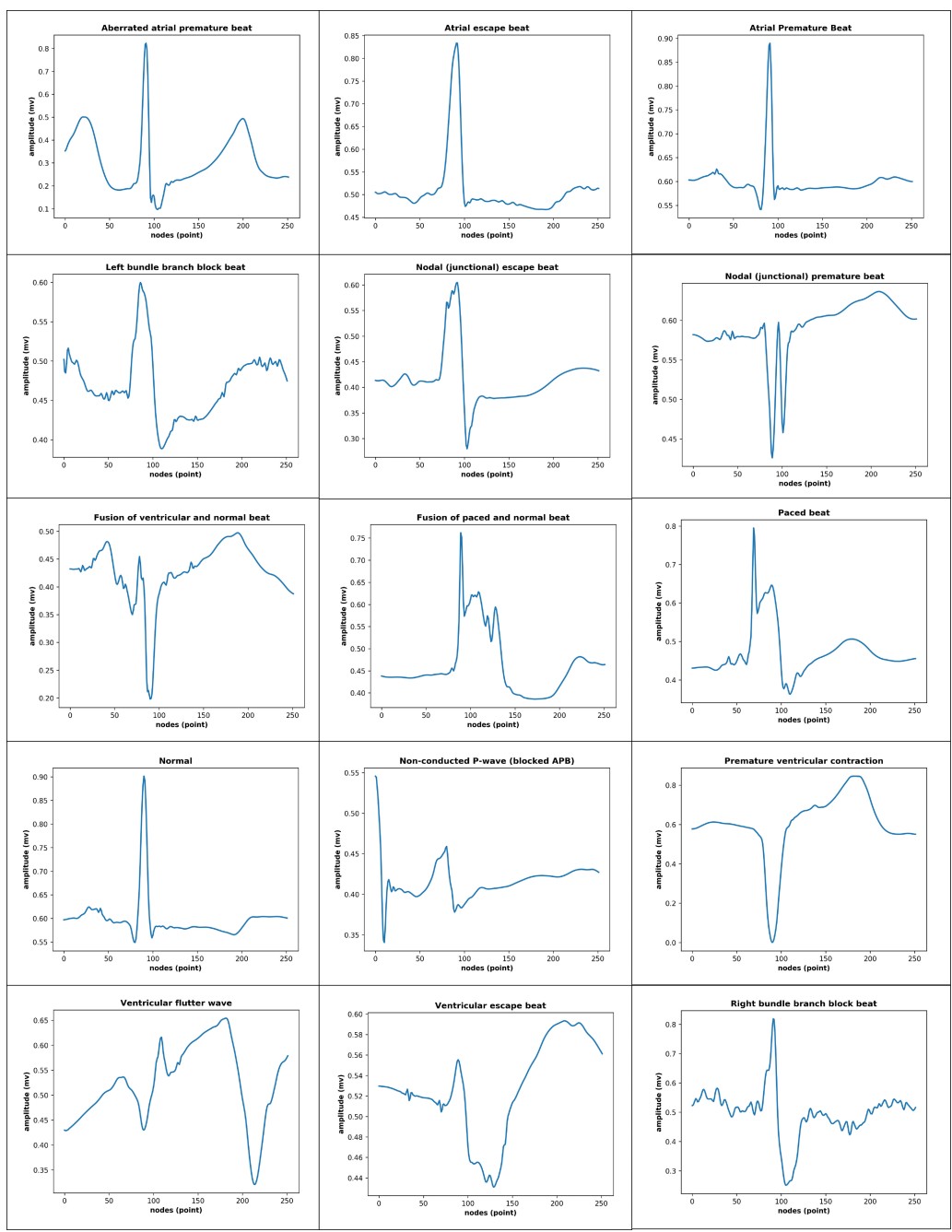

**Figure 2 15-class of ECG-based beat classification.**

- ECG beat segmentation is the step of intercepting numerous nodes in a signal to discern not only subsequent heart beats but also the waveforms included in each beat (*Qin et al., 2017*). The former refers to the characteristics retrieved from a single beat, which typically only contains one R-peak. The latter, however, refers to features that are dependent on at least two beats. These features include more information

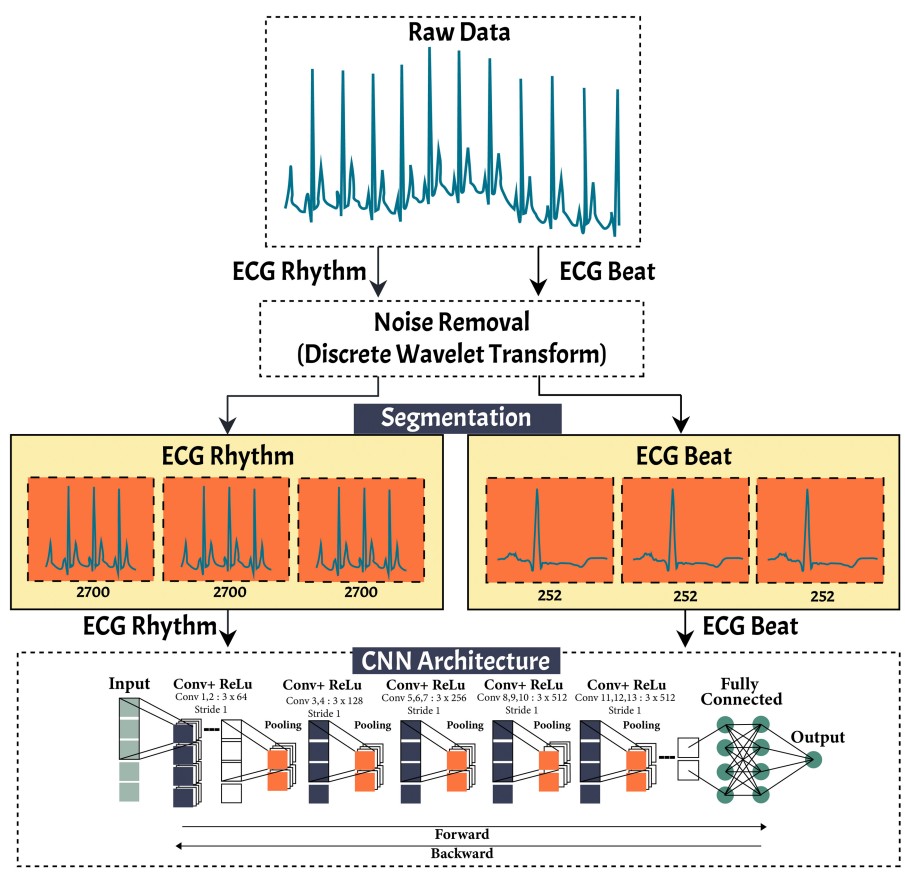

**Figure 3** **The proposed research methodology of ECG rhythm and beat classification.**

than a single R-peak. The waveforms of beat segmentation are presented in Fig. 4B. Figure 4B shows the positions of the P-wave, QRS-complex, and T-wave, which are all intimately connected to the location of the R-peak. According to *Qin et al. (2017)*, *Chang et al. (2012)* and *Nurmaini et al. (2019)*, the average ECG rhythm frequency is between 60 and 80 beats per minute, the t1 duration is 0.25 s before R-peak, and the t2 duration is 0.45 s after R-peak, which results in a total length of 0.7 s. A total of 0.7 s contains 252 nodes, with a sampling frequency of 360 Hz, which covers the P-wave, QRS-complex, and T-wave (one beat).

4. **Feature Extraction and Classification** The 1D-CNN classifier was proposed by *Nurmaini et al. (2020)* for AF detection. By using the architecture, we generalized the model for abnormal–normal rhythm and beat classification. The rectified linear unit (ReLU) function was adopted with 13 convolution layers (64, 128, 256, and 512 filters) and also consisted of five max pooling layers. The 1D-CNN model comprised two fully connected layers with 1,000 nodes for each layer and one node for the output layer. The 1D-CNN required a three-dimensional input, which consists of *n* samples, *n* features, and timesteps. The detailed process of 1D-CNN architecture for both ECG rhythm and beat classification was as follows:

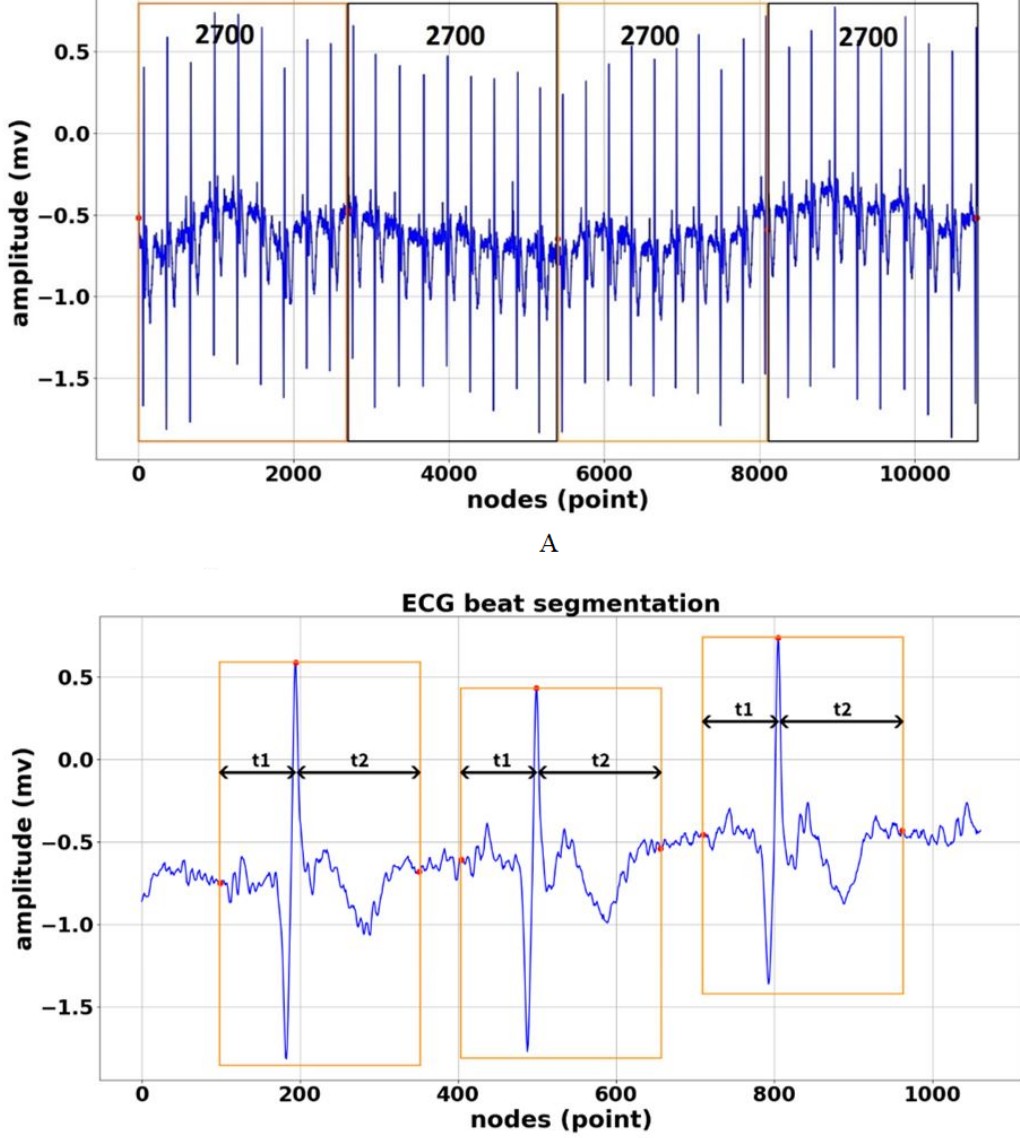

**Figure 4** ECG signal segmentation (A) Segmented in 2700 nodes for rhythm feature (B) Segmented in 252 nodes for beat feature.

- For ECG based on rhythm classification, the input timesteps with the dimension 2700 ×1 were fed into the convolution layer equipped with the ReLU activation function. The first and second convolutional layers produced an output length of 64 with a kernel size of 3. The output of the first and second convolutional layers through the max pooling layer had a kernel size of 2 for the feature reduction. The output of the first max pooling layer as the input for the third and fourth convolutional layers produced 128 feature maps. The convolutional layers were

passed onto the fifth and last convolutional layers and produced output lengths of 256 and 512, respectively, with a kernel size of 3. The output of the last convolutional layer was then passed onto two fully connected layers with a total of 1,000 nodes. This architecture produced an output of a nine-class ECG rhythm classification.

- Unlike ECG rhythm classification, none of the processes differed from the features interpretation for ECG based on beat classification. The main differences were the input timesteps value of (252, 1) and products of the output size of the 15-class ECG beat classification. The architecture also implemented the ReLU activation function with 64, 128, 256, and 512 filters, with a kernel size of 3. For each max pooling layer, a kernel size of 2 was also used for the feature interpretation of the 1D-CNN for ECG beat classification.

5. **Model Evaluation** Classification ECG signal based on rhythm and beat feature is evaluated by using intra and inter patient scheme. Such schemes are conducted to resemble a clinical environment and to ensure the robustness of the proposed model. Five commons metrics used in this study are accuracy, sensitivity, specificity, precision and F1-score. Moreover, two measures are usually considered for evaluating the classification performance, specifically for imbalance data, are receiver-operating characteristic (ROC) and Precision-Recall (P-R) curves. These two-evaluation metrics were added because the overall accuracy was distorted by the majority class results, since the beat type classes are extremely imbalanced in the available dataset.

## RESULTS AND DISCUSSION

For ECG rhythm and beat classification, the proposed 1D-CNN model was tested on an validation set (intra-patient) and unseen set (inter-patient) in this study. All experimentation in the training processes used a 10-fold cross-validation scheme. This scheme divides the collection of observations into $k$ groups, or folds, of roughly similar size at random. The scheme is fitted on the remaining $k - 1$ folds, with the initial fold serving as a validation set. For the selected model, the parameters that provided the best cross-validation accuracy, sensitivity, specificity, precision, and F1 score were chosen.

### ECG rhythm classification in validation model

A total of 2,445 records consisted of rhythm episodes of 138,415 training sets, 15,373 validation sets, and 14,684 unseen sets after being segmented by each 2,700 nodes (refer to Table 3). A total of 168,472 episodes were analyzed for the ECG rhythm classification task. As can be seen, all PTB Diagnostics ECG records were used for the training and validation sets. The rest of the datasets were used for the training, validation, and unseen sets. Table 3 shows the large different ratio between one class and another (imbalanced) class, for example, a total number of MI, HF, and HC classes to H, M, and VHD classes. However, we did not implement the oversampling techniques to overcome such a case in this study.

Without considering the data ratio (total number) of episodes for each class, we validated the proposed 1D-CNN model to the 10-fold cross-validation scheme. Figure 5

**Table 3  The total episodes after segmentation of 2700 nodes.**

| Dataset | Class | Total rhythm after segmentation of 2700 nodes (episode) | | Unseen set |
| --- | --- | --- | --- | --- |
| | | Training set | Validation set | |
| PTB diagnostics ECG | BBB | 230 | 26 | |
| | C | 658 | 73 | |
| | D | 619 | 69 | |
| | HC | 3,096 | 344 | |
| | H | 271 | 30 | – |
| | MI | 14,242 | 1,582 | |
| | M | 155 | 17 | |
| | VHD | 232 | 26 | |
| BIDMC congestive heart failures | HF | 53,738 | 5,969 | 6,647 |
| China physiological signal challenge 2018 | BBB | 4,651 | 514 | 614 |
| | BBB | | | |
| MIT-BIH normal sinus rhythm | HC | 60,523 | 6,723 | 7,423 |
| Total | | 138,415 | 15,373 | 14,684 |

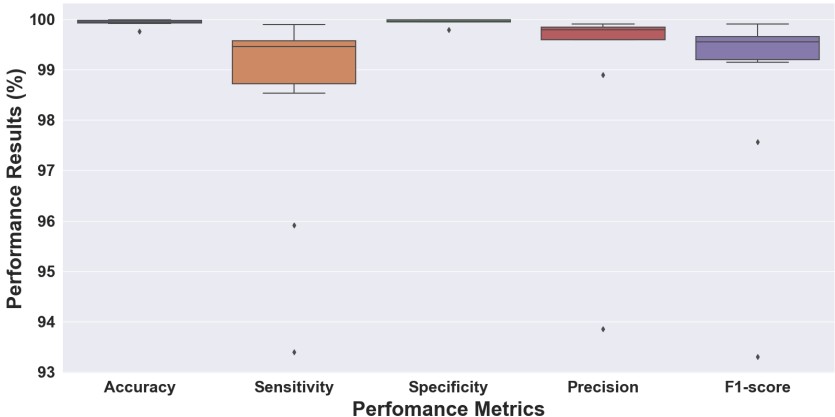

**Figure 5  Boxplot of the 10-fold cross-validation results for ECG rhythm classification.**

shows the performance results of folds 1 through 10, which were evaluated for accuracy, sensitivity, specificity, precision, and F1 score. The performance results obtained above 99% for accuracy and specificity and ranged from 93 to 99% for sensitivity, precision, and F1 score. The model with the highest accuracy was chosen as the best model for this study out of all the models analyzed. The model had an accuracy of 99.98%, a sensitivity of 98.53%, a specificity of 99.99%, a precision of 99.81%, and an F1 score of 99.15% (fold 6). The results showed that the proposed 1D-CNN was the most accurate predictor, with an accuracy of 99.98%.

Confusion Matrix (CM) for rhythm evaluation result is shown in Fig. 6. This metric is used to capture information about the predicted results from the model respected to the actual label. It can be seen that the BBB class has four prediction errors (predicted as

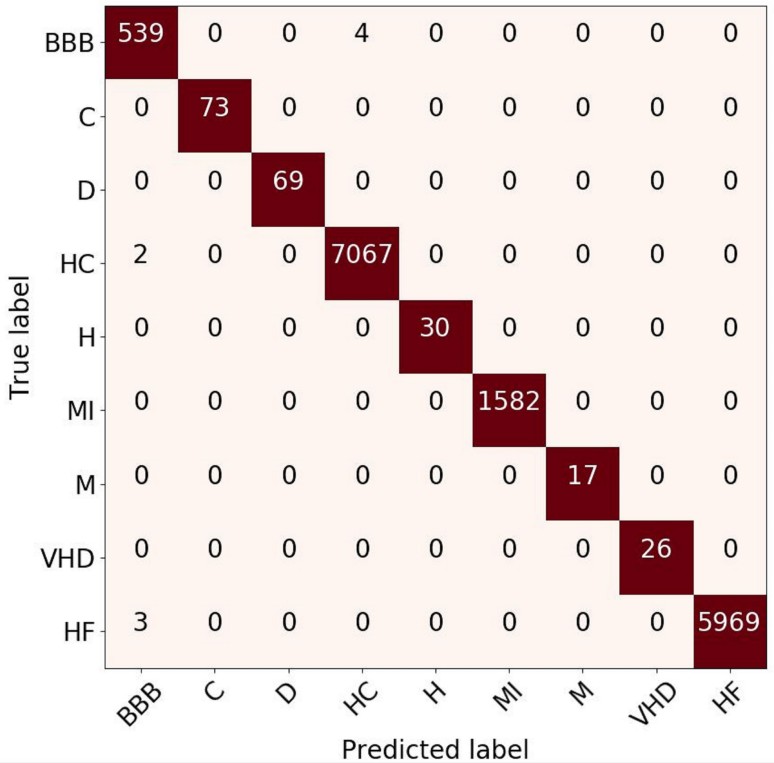

**Figure 6  Confusion matrix evaluation for ECG rhythm classification on validation result.**

healthy-control class) and two healthy-control rhythms that are predicted as BBB. The prediction error between those arises due to the morphology of these types of rhythms is almost similar. Even so, the overall predictive results of the proposed approach provide a satisfactory evaluation performance. Based on the CM, the classification result of the ECG signal with rhythm feature produces good performance, due to only two class of ECG pattern have misclassified (HC and HF). However, overall result can be state that the classification is close to 100%.

Using the performance value in CM, we can observe the classification result with other views in terms of classification model at all classification thresholds named receiver-operating characteristic (ROC) and precision–recall (PR) curves. This curve plots two parameters true positive rate (sensitivity) and false positive rate (specificity) provide a graphical representation of a classifier's performance across many thresholds, rather than a single value. It is important to understand the trade-off in performance for different threshold values. As shown in Figs. 7A and 7B. Figure 7A shows the resulting ROC curve, which compares the nine-class of ECG rhythm characteristics. The comparable value is sensitivity *versus* specificity. Sensitivity is the ability to correctly identify the true positive class of ECG rhythm, whereas specificity is the ability to correctly identify the true negative rate of ECG rhythm. Therefore, if used in medical data, it will produce a precise and

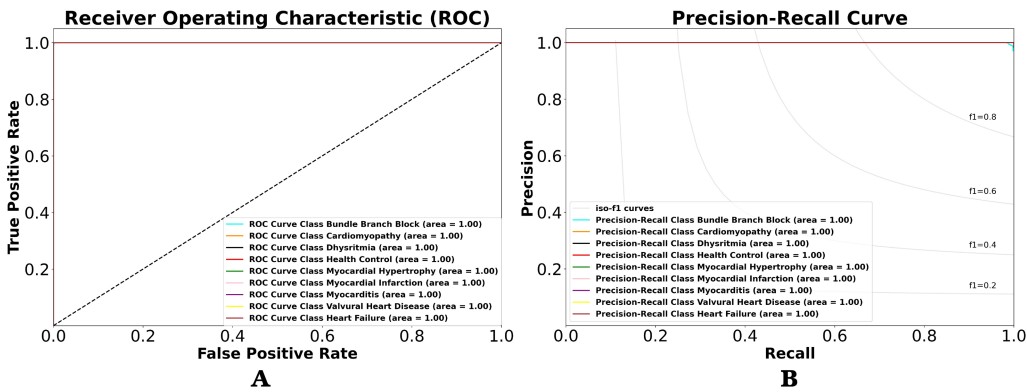

**Figure 7** (A) ROC and (B) P-R curves for ECG rhythm classification on the validation result.

accurate diagnosis. Misclassification between positive class and negative class of ECG rhythm can be dangerous, and the consequences can be as serious as death.

The area under the curve (AUC) is the value analyzed in the ROC by looking at how far the middle value is and whether the area below the curve approaches the value of 1. The lower left point of the graph (0,0) is a value that does not contain errors (no false positives) and does not detect any true positives. On the upper right side of the graph (1,1), the opposite point defines all true positives but with a 100% error rate (rates of false positives). The upper left point (0,1) is the ideal classification that defines all true positives without any mistakes (no false positives or 0 cost). The lower right point (1,0) is the worst classification, where all subjects labeled as positive are simply false positives, without knowing true positives. As shown in Fig. 7A, the ROCs of the nine-class normal-abnormal ECG rhythm show excellent performance, as the value of the ROC for the nine-class classification is 1, or the AUC is about 100%. This means that the proposed 1D-CNN can categorize all classes with higher accuracy and precision. However, the ROC cannot be trusted with imbalanced data, and it remains unchanged even after the performance changes. Therefore, the P-R curve is used to describe the classifier performance on imbalanced data (Fig. 7B). The overall performances are also good, as the P-R value is 1.

Table 4 lists the performance results for the nine-class of ECG based on rhythm feature in the intra-patient scheme. As can be seen, the C, D, H, MI, M, and VHD classes obtained 100% for accuracy, sensitivity, specificity, precision, and F1 score. The proposed 1D-CNN model was proven to be robust and had no effect on the imbalanced class problem. For the nine-class classification, the average of all performance metrics achieved above 99% accuracy.

## ECG beat classification in validation model

For the 15-class of ECG beats, a total of 110,082 beats were trained, validated, and tested (unseen) in this study. The large different ratio between one class and another (imbalanced) class, however in this study we can't conducted the augmentation data. All ECG beats data divided into a ratio of 8: 2 or 80% is used for training data and the remaining for testing.

**Table 4   Performance results of the nine class with ECG rhythm in the intra-patient scheme.**

| Performance metrics (%) | Class | | | | | | | | | Average |
|---|---|---|---|---|---|---|---|---|---|---|
| | BBB | C | D | HC | H | MI | M | VHD | HF | |
| Accuracy | 99.94 | 100 | 100 | 99.96 | 100 | 100 | 100 | 100 | 99.98 | 99.98 |
| Sensitivity | 99.26 | 100 | 100 | 99.97 | 100 | 100 | 100 | 100 | 99.95 | 99.90 |
| Specificity | 99.08 | 100 | 100 | 99.94 | 100 | 100 | 100 | 100 | 100.00 | 99.89 |
| Precision | 99.17 | 100 | 100 | 99.96 | 100 | 100 | 100 | 100 | 99.97 | 99.90 |
| F1-Score | 99.97 | 100 | 100 | 99.98 | 100 | 100 | 100 | 100 | 99.97 | 99.99 |

The process of training with 10-fold is selected with randomly. Therefore, approximately 88,065 beats are used as training data and about 22,017 beats as testing data.

The performance results of folds 1 through 10 using the 10-fold cross-validation scheme are shown in Fig. 8. As can be seen, the results vary from 0% as the lowest and around 99% as the highest result. Accuracy and specificity achieved above 99%, sensitivity and precision ranged from above 0% to 94%, and the F1 score ranged from 0% to 94%. The model had an accuracy of 99.88%, a sensitivity of 96.98%, a specificity of 99.90%, a precision of 92.24%, and an F1 score of 94.39% (fold 6). Unlike the ECG rhythm results, the performance of the 10-fold was not good enough. There was an outlier of sensitivity, which had a 0 (zero) value in the initial fold. The massive difference between the total number of the normal beat (N) class and the other abnormal beats could be an imbalanced class problem.

To analyze the performance of the 15-class of ECG beats, we also presented the confusion matrix evaluation in Fig. 9. It can be seen that normal beats have the highest number of true positives (with 7,248 data). However, this beat also has the most false-negative and false positive values with 46 and 21 data, respectively compared to other classes. Furthermore, both classes J and e have neither false positive nor false negative errors. Even though the ratio of number data used in this study was imbalance, the atrial escape beat (e), which is proven to be a minority class, was able to classify by the model correctly. However, an imbalanced data problem still requires a particular concern to avoid the model simply predicting the majority class rather than the minority.

To analyze the performance of the 15-class of ECG beats, we also presented the ROC and P-R curve (refer to Figs. 10A and 10B). Figure 10A shows that the perfect classification can be presented in the R, L, j, and P beat classes. However, the other beat classes obtained an AUC value above 75%. Also, Fig. 10B shows the worst classification as the e beat class, with an AUC value above 50%. According to the ratio of number data that was used in this study, the atrial escape beat (e) is proven to be a minority class, as it has limited dataset representation. Due to the large imbalanced class, the model tends to perform poorly and requires some modifications to avoid simply predicting the majority class in all cases.

The results for the 15-class ECG beat classification are listed in Table 5. As can be seen, the results show an above 99% accuracy and specificity for all 15-class of ECG beats. The results presentation is quite good for the ECG beat classification task, although some beats' results (A, a, and j) are not. The A and a beats are related to an atrial premature beat, causing aberrant ventricular conduction. An unexpected beat discharged by an ectopic

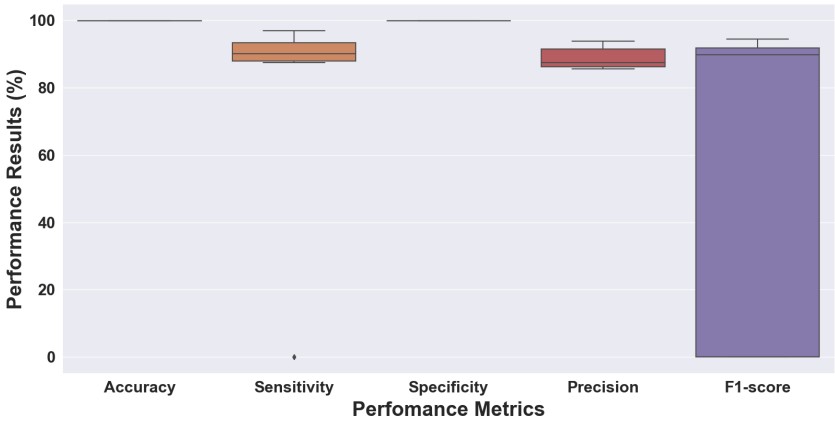

**Figure 8** Boxplot of the 10-fold cross-validation results for the ECG beat classifiation.

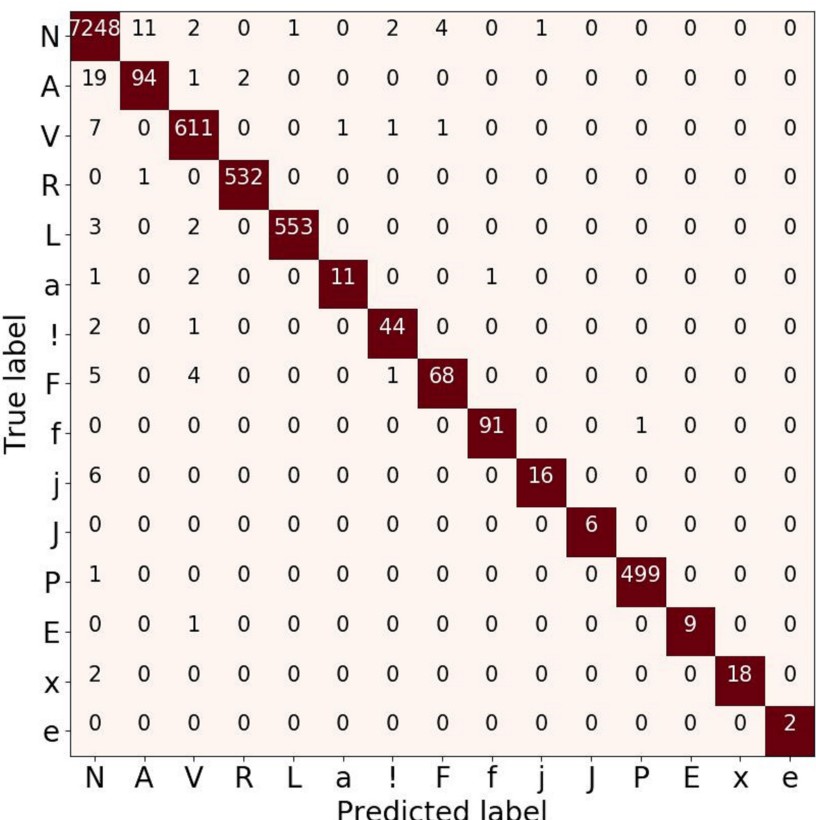

**Figure 9** Confusion matrix evaluation for ECG beat classification on validation result.

focus in the atria is termed a premature atrial beat. While some fibers are still refractory, the impulse from the premature beat reaches the His-Purkinje system early. Due to abnormal ventricular conduction, the resultant QRS complex exhibits a right BBB pattern. Also,

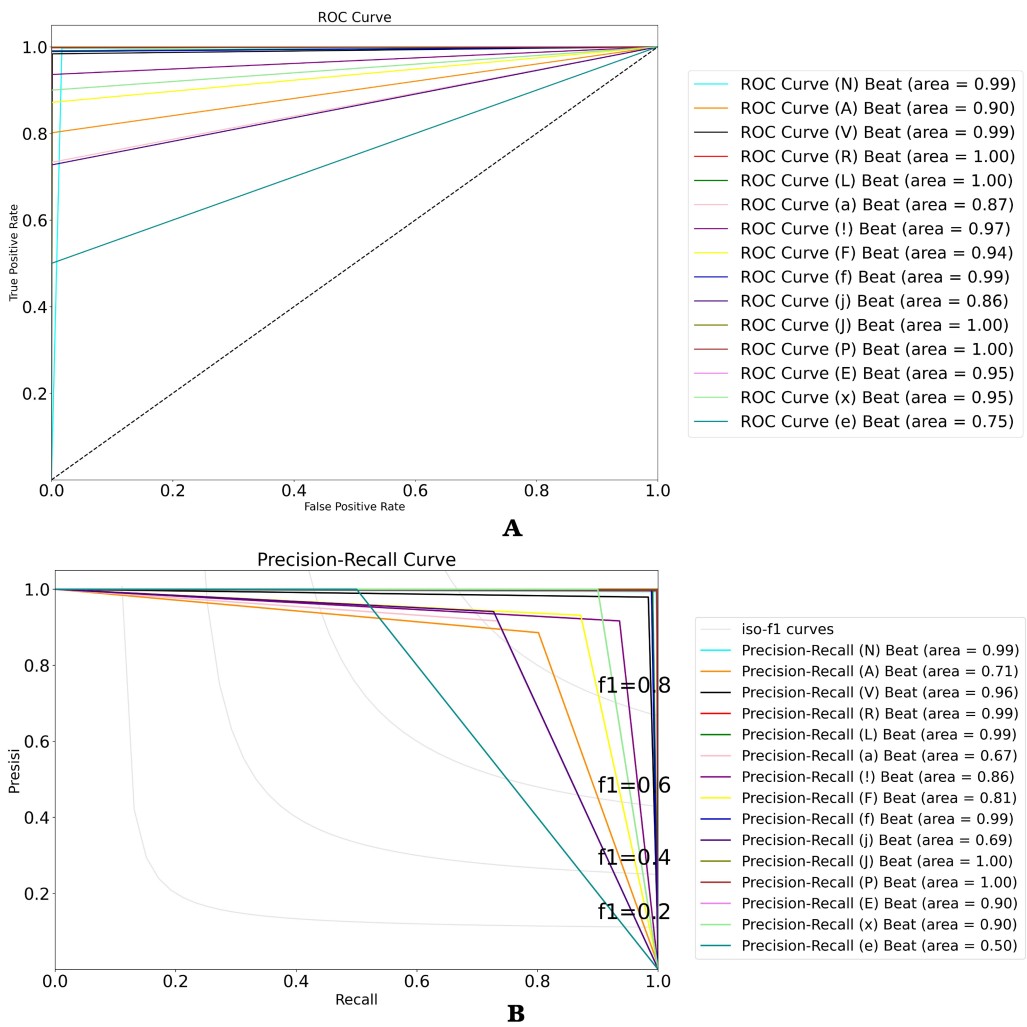

**Figure 10 (A) ROC and (B) P-R curves for ECG beat classification on the validation result.**

the j beat is a delayed heartbeat originating from an ectopic focus in the atrioventricular junction. The classification of ECG beats tends to be more challenging because the results are related to the heart beat segmentation process, which will be close to optimal with the QRS detection.

## ECG signal classification with inter-patient data

Tables 4 and 5 list the proposed model result with dataset based on intra-patient scenario. Such conditions where the ECG data from the same patients probably appear in the training and validation set. In this study, we took the precaution to construct and evaluate the classification using rhythm and beat features also from different patients (inter-patient). To test the robustness of the proposed 1D-CNN model, we tested the model on an unseen set (refer to Table 6). The unseen set sample consisted of five of the 24-class of ECG-based

**Table 5** The performance results of the 15-class with ECG beats in the intra-patient scheme.

| Beats class | Performance results (%) | | | | |
|---|---|---|---|---|---|
| | Accuracy | Senisitivity | Specificity | Precision | F1-Score |
| N | 99.32 | 99.36 | 99.19 | 99.71 | 99.53 |
| A | 99.65 | 88.67 | 99.77 | 81.03 | 84.68 |
| V | 99.76 | 97.91 | 99.89 | 98.38 | 98.15 |
| R | 99.96 | 99.62 | 99.98 | 99.81 | 99.71 |
| L | 99.93 | 99.81 | 99.94 | 99.10 | 99.46 |
| a | 99.94 | 91.66 | 99.95 | 73.33 | 81.48 |
| ! | 99.92 | 91.66 | 99.96 | 93.61 | 92.63 |
| F | 99.84 | 93.15 | 99.89 | 87.17 | 90.06 |
| f | 99.97 | 98.91 | 99.98 | 98.91 | 98.91 |
| j | 99.92 | 94.11 | 99.93 | 72.72 | 82.05 |
| J | 100 | 100 | 100 | 100 | 100 |
| P | 99.97 | 99.8 | 99.98 | 99.8 | 99.8 |
| E | 99.98 | 100 | 99.98 | 90 | 94.73 |
| x | 99.97 | 100 | 99.97 | 90 | 94.73 |
| e | 100 | 100 | 100 | 100 | 100 |
| Average | 99.87 | 96.97 | 99.89 | 92.23 | 94.39 |

**Table 6** Performance results of the inter-patient scheme.

| Performance metrics (%) | Class | | | | |
|---|---|---|---|---|---|
| | BBB | HC | HF | V | L |
| Accuracy | 99.97 | 99.99 | 99.98 | 97.05 | 99.53 |
| Sensitivity | 98.91 | 100.0 | 100.0 | 96.19 | 99.00 |
| Specificity | 100.0 | 99.98 | 99.97 | 97.13 | 99.68 |
| Precision | 99.45 | 99.99 | 99.98 | 84.98 | 98.88 |
| F1-Score | 99.97 | 100.0 | 100.0 | 84.27 | 98.94 |

rhythms and beat feature—BBB, HC, HF, V and L class. From the experiment, the performance still achieved outstanding results.

## Benchmarking of the proposed model

The comparison results of our proposed 1D-CNN architecture with the state-of-the-art model are listed in Table 7. In order to make a fair benchmarking, we compare our proposed model with several previous studies. These studies focus on the ECG signal classification using DL architecture, especially the use of 1D-CNN architecture (*Yıldırım, Pławiak & Rajendra Acharya, 2018*; *Rajkumar, Ganesan & Lavanya, 2019*; *Nannavecchia et al., 2021*), LSTM architecture (*Yildirim et al., 2019*; *Gao et al., 2019*), and combination architecture of 1D-CNN as a feature extraction and LSTM as a classifier (*Lui & Chow, 2018*; *Oh et al., 2018*; *Yildirim et al., 2020*; *Chen et al., 2020*; *Luo et al., 2021*). However, all the classification methodologies are developed by treating beat and rhythm separately. In contrast, our study utilizes a single architecture based on 1D-CNN architecture through both features,

**Table 7  Comparison results with the state of the art.**

| Authors | Class | Feature | Method | Performance results (%) | | | |
|---|---|---|---|---|---|---|---|
| | | | | Acc. | Sens. | Spec. | Pre. |
| *Rajkumar, Ganesan & Lavanya (2019)* | 8 | rhythm | 1D-CNN | 93.60 | – | – | – |
| *Yıldırım, Pławiak & Rajendra Acharya (2018)* | 17 | rhythm | 1D-CNN | 91.30 | 83.90 | – | 85.4 |
| *Nannavecchia et al. (2021)* | 21 | beat | 1D-CNN | 89.51 | 87.79 | – | 86.78 |
| *Yildirim et al. (2019)* | 5 | rhythm | LSTM | 99.23 | – | – | 99.00 |
| *Gao et al. (2019)* | 8 | rhythm | LSTM | 99.26 | – | 99.26 | 99.14 |
| *Lui & Chow (2018)* | 4 | beat | 1D-CNN-LSTM | – | 92.40 | 97.70 | – |
| *Yildirim et al. (2020)* | 7 | beat | 1D-CNN-LSTM | 92.24 | 80.15 | 98.72 | 80.31 |
| *Oh et al. (2018)* | 5 | rhythm | 1D-CNN-LSTM | 98.10 | – | – | 97.50 |
| *Chen et al. (2020)* | 6 | rhythm | 1D-CNN-LSTM | 99.32 | 97.75 | – | – |
| *Luo et al. (2021)* | 9 | rhythm | 1D-CNN-LSTM-GRU | 99.01 | 99.58 | – | 99.44 |
| Our work | 9 | beat | 1D-CNN | 99.98 | 99.90 | 99.89 | 99.90 |
| | 15 | rhythm | 1D-CNN | 99.87 | 96.97 | 99.89 | 92.23 |

**Notes.**
  *Acc. (accuracy); Sen. (sensitivity); Spec. (specificity); Pre, precision.

rhythm and beat, to classify 24 patterns of ECG signals. In the ECG signal interpretation, the abnormalities can be analyzed using heart beat or heart rhythm feature. Therefore, such process is more efficient to integrate two features for classifying ECG signals in one architecture. To our knowledge, no studies developed such combination scenario with one architecture. Thus, in this study, we will analyze and compare the classification results for beat and rhythm features separately.

It can be seen in Table 7, the previous studies show that the ECG signal classification based on rhythm feature propose 1D-CNN architecture for 17 classes with 91.30% accuracy (*Yıldırım, Pławiak & Rajendra Acharya, 2018*), and propose LSTM architecture for five classes with 99.23% accuracy (*Yildirim et al., 2019*). From these two studies, the classification performance is improved; however, the number of classes are reduced from 17 to 5-class. Other study for ECG signal classification based on beat feature utilize combination between 1D convolutional layers and LSTM architecture with 10.000 subject and seven-class abnormalities (*Yildirim et al., 2020*). By using the proposed model, the classification accuracy around 92.24%, unfortunately, the sensitivity only reaches 80.15%. It means that the smallest change in the ECG signal can't be detected by the network. They use convolutional layers to produce a low- and high-level feature, however, the LSTM classifiers lack to recognize the dynamic of ECG feature extracted from CNN. The sensitivity is important value in medical analysis, its relation to the number of false-negative result. The small sensitivity value indicates that many ECG signals are misclassified. Such case also occurred in *Lui & Chow (2018)*, they use 1D-CNN-LSTM architecture, however the sensitivity only reaches 92.40%.

Combination of 1D Convolutional layers with other DL architecture can actually produce quite impressive results (*Lui & Chow, 2018*; *Oh et al., 2018*; *Chen et al., 2020*; *Yildirim et al., 2020*; *Luo et al., 2021*) rivalling a model which uses individual 1D-CNN and LSTM architecture (*Yıldırım, Pławiak & Rajendra Acharya, 2018*; *Gao et al., 2019*;

*Rajkumar, Ganesan & Lavanya, 2019*; *Nannavecchia et al., 2021*). Even in other study, in order to obtain satisfactory results in ECG signal classification, three different DL architecture, 1D-CNN, LSTM and GRU are combined (*Luo et al., 2021*). Unfortunately, they used SMOTE algorithm to eliminate the imbalanced problems. By using resampling methods can increase the overlapping between classes and can introduce additional noise. The beat classification is still challenging, when the number of classes is increased it will decline the performance. In *Nannavecchia et al. (2021)*, they classify ECG signal for 21 classes abnormalities. However, the classification result is unsatisfactory with 89.51% accuracy and 87.79% sensitivity, which means a large of number classes are misclassified.

In our proposed model only utilize 1D-CNN architecture with 13 convolutional layers and five max-pooling layers, we produce a satisfactory result using both beat and rhythm features. Our model performance outperforms other studies with a large number of classes. All the performance of nine-class classification value reach over 99% by using beats feature, but the classification sensitivity decreases to 96.67% by using rhythm feature. It happened because we use 15 classes of ECG signal abnormalities with an unbalanced number of classes. Such condition causes the classifiers tend to make biased learning model that has a poorer predictive accuracy over the minority classes compared to the majority classes. Even though, our model still maintains the classification performance with imbalanced data, the sensitivity value is decreased only 3%, it does not affect the performance significantly. One dimensional convolutional learning methods are more efficient to learn local patterns than a recurrent neural network. It contains a sliding filter, which may be regarded as moving across the input by sharing weights over a local patch function. It concludes that 1D-CNN performs better in areas where local patterns are important for classification task.

Although the results look promising for ECG rhythm and beat classification, there are some limitations to our study:

- The pre-processing stage of the ECG signal still needs improvement, specifically in the case of ECG signals that have different sampling frequencies, leads, and various noises.
- The segmentation of the P, QRS, and T-waves and the HRV measurement before the classification process were not carried out; and
- The proposed model was not validated against the hospital patient data. We only used the available public dataset.

## CONCLUSIONS

Deep learning has gained a central position in recent years for ECG rhythm and beat classification. It was built on a foundation of significant algorithmic details and generally can be understood in the construction and training of DL architectures. A DL approach based on one 1D-CNN architecture has been presented to automatically learn and classify the nine-class of ECG pattern with rhythms feature and 15-class of ECG pattern with beats feature, which is important for classifying the abnormalities pattern. In this study, the proposed 1D-CNN model, which consisted of 13 convolutional layers and five max-pooling layers, was used. The 1D-CNN has low computational requirements. Thus, it is well-suited for real-time and low-cost applications for ECG devices.

Using the 10-fold cross-validation scheme, the performance results had an accuracy of 99.98%, a sensitivity of 99.90%, a specificity of 99.89%, a precision of 99.90%, and an F1 score of 99.99% for ECG rhythm classification. Also, for ECG beat classification, the model obtained an accuracy of 99.87%, a sensitivity of 96.97%, a specificity of 99.89%, a precision of 92.23%, and an F1 score of 94.39%. We realize the performance results of the ECG rhythm are better than the ECG beat classification. The selection of an appropriate preprocessing step for QRS detection to accurately find the R-peak to achieve the best model for ECG beat classification is needed to achieve high performance results. In the future, the challenges regarding ECG signals are still many, such as the precision segmentation of P, QRS, and T-waves before the process of rhythm and beat classification.

### Funding
This work was supported by Intelligent System Research Group (ISysRG), Universitas Sriwijaya, Indonesia. The funders had no role in study design, data collection and analysis, decision to publish, or preparation of the manuscript.

### Grant Disclosures
The following grant information was disclosed by the authors:
Intelligent System Research Group (ISysRG), Universitas Sriwijaya, Indonesia.

### Competing Interests
The authors declare there are no competing interests.

### Author Contributions
- Annisa Darmawahyuni conceived and designed the experiments, performed the experiments, analyzed the data, prepared figures and/or tables, authored or reviewed drafts of the paper, and approved the final draft.
- Siti Nurmaini conceived and designed the experiments, authored or reviewed drafts of the paper, and approved the final draft.
- Muhammad Naufal Rachmatullah and Firdaus Firdaus analyzed the data, performed the computation work, authored or reviewed drafts of the paper, and approved the final draft.
- Bambang Tutuko performed the experiments, analyzed the data, authored or reviewed drafts of the paper, and approved the final draft.
- Ade Iriani Sapitri analyzed the data, prepared figures and/or tables, and approved the final draft.
- Ahmad Fansyuri and Aldi Predyansyah performed the experiments, performed the computation work, prepared figures and/or tables, and approved the final draft.

### Data Availability
The code is available in the Supplemental File.

The raw data are available at:

1. 2018 China Challenge dataset: http://2018.icbeb.org/Challenge.html

2. PTB Diagnostic ECG Database: https://www.physionet.org/content/ptbdb/1.0.0/

3. BIDMC Congestive Heart Failure Database: https://www.physionet.org/content/chfdb/1.0.0/

4. MIT-BIH Arrhythmia Database: https://www.physionet.org/content/mitdb/1.0.0/

5. MIT-BIH Normal Sinus Rhythm Database: https://www.physionet.org/content/nsrdb/1.0.0/

## Supplemental Information

Supplemental information for this article can be found online at http://dx.doi.org/10.7717/peerj-cs.825#supplemental-information.

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
