# Peer review of "Deep learning-based electrocardiogram rhythm and beat features for heart abnormality classification"

_PeerJ Computer Science, doi:10.7717/peerj-cs.825_

## Round 0.1 · original submission · Major Revisions

The article has some good content but the reviewers highlighted some important issues that need to be solved. Please address the points raised by the reviewers and prepare a new version.

In the binary classification results section, please include the results measured through the Matthews correlation coefficient (MCC).

Reviewer 1 ·

Basic reporting

The manuscript is well-written, has good presentation, and thoroughly covers the literature.

Experimental design

The manuscript should be clearer in stating the novelty of the proposed method versus the existing state-of-the-art. Also, the architecture of the proposed network should be more clearly presented.

Validity of the findings

Results seem good, and the experiments use plenty of databases. However, the comparison with the state-of-the-art is flawed and does not suffice to assess the relative quality of the proposed method. The most promising literature approaches should be tested in the same conditions as the proposed method to directly compare their performance results.

Additional comments

The exact contribution of this work is not very clear. Throughout the abstract and introduction, the authors state that the literature is composed of either rhythm or beat-based methods, which leads me to think that this would be a hybrid method combining rhythm and beat features. However, both tasks are addressed separately, which was already studied in (Nurmaini 2020) and (Tutuko 2021). The introduction should state, more clearly, what is the difference between this and the previous works.

The figure 3, depicting the CNN architecture, needs to be improved. As it stands, it seems there is a FC neuron (in the bottom) for each convolutional layer, and somehow information flows forward and backward through the model using those neurons. Also, it is not clear what the orange squares represent.

Were the state-of-the-art methods (in Table 7) evaluated with the same datasets as the proposed method? And the same data train-test splits? If not, and since they even consider different number of target classes, the results are not really comparable. Hence, we do not have a robust way to assess if the proposed method is, in fact, better than the state-of-the-art. Authors should implement a couple of the most promising literature methods and test them in the same conditions as the proposed method.

In figure 6, since the AUC is very close to 1, perhaps it would be useful to use log-log scale axes for the ROC curve. Or, there could be a box zooming in on the [1.0, 1.0] area, so readers can see clearly the difference between the curves.

·

Basic reporting

In this manuscript, Darmawahyuni and colleagues evaluated the performance of 1D-CNN model on ECG classification. The authors reported that their model demonstrated high classification performance on both rhythm and beat abnormalities. The reviewer agrees with the clinical importance of simultaneously identifying ECG waveforms and rhythms, but unfortunately the authors' method fails to achieve their goal.

The authors stated that they were able to classify 24 heart abnormalities (9 rhythms and 15 beats), but in reality, they only repeated the binary classification 24 times. This is the main drawback of this study. As a side note, the reviewer believes that confusion matrices, not ROC curves, should be used to evaluate the performance of multiclass classification.

The definition of the classifications for cardiac rhythm is also incorrect. According to the authors, the rhythm abnormality includes “Myocardial Infarction”, Myocarditis”, “Heart Failure”, “Valvular heart disease”, “Hypertrophy” and “Cardiomyopathy”, none of these are abnormalities in heart rhythm. Furthermore, since myocardial infarction and heart failure often coexist, and valvular disease and cardiac hypertrophy can also coexist, classifying them as separate classes (based on ECG alone) is obviously a wrong approach. In such a complex task, it is impossible for all classification models to achieve AUROC 1.0, as shown in Figure 6.

The reviewers speculate that these mistakes may be partly due to the lack of clinicians among the authors. This manuscript needs to be radically revised from the research design.

Experimental design

Experimental design is inappropriate as described above.

Validity of the findings

Findings are unreliable as described above.

---

## Round 0.2 · Major Revisions

As the reviewer noticed, a direct comparison with the state-of-the-art is missing. The authors must include it if they want to reach the acceptance of this article.

Reviewer 1 ·

Basic reporting

The manuscript is well-written, has a good presentation, and thoroughly covers the literature.

Experimental design

The experiments have been correctly and thoroughly defined and conducted.

Validity of the findings

Results seem good, and the experiments use plenty of databases. The comparison with the state-of-the-art remains flawed and does not suffice to assess the relative quality of the proposed method. The most promising literature approaches should be tested in the exact same conditions as the proposed method to directly compare their performance results.

Additional comments

I thank the authors for their response to my comments. I believe most have been addressed, although one key concern has been avoided (and the manuscript's quality has suffered because of that).

Direct comparison with the state-of-the-art is fundamental. If you have more classes than the literature, or other things are different (random seeds, etc.), then implement state-of-the-art methods and test them in your exact scenario. If there are many competing works in the literature, select at least the 1-2 best to have this direct benchmarking. This is essential to have a real fair comparison and to really understand if, where, and how your method is best.

---

## Round 0.3 · accepted · Accept

The authors addressed the main concerns of the reviewer so we can now recommend this article for publication.

Reviewer 1 ·

Basic reporting

The manuscript is well-written, has a good presentation, and thoroughly covers the literature.

Experimental design

The experiments have been correctly and thoroughly defined and conducted.

Validity of the findings

Results seem good, and the experiments use plenty of databases. The manuscript now includes direct comparison with state-of-the-art methods to objectively illustrate the superiority of the proposed method.

Additional comments

I thank the authors for addressing my comments. I believe it is now publishable. Congratulations on a good piece of research!